# The Feasibility of Pressurised Intraperitoneal Aerosolised Virotherapy (PIPAV) to Administer Oncolytic Adenoviruses

**DOI:** 10.3390/pharmaceutics13122043

**Published:** 2021-11-30

**Authors:** Sophia J. Tate, Leen Van de Sande, Wim P. Ceelen, Jared Torkington, Alan L. Parker

**Affiliations:** 1Division of Cancer and Genetics, Cardiff University, Cardiff CF14 4XN, UK; TateSJ@Cardiff.ac.uk; 2Department of Human Structure and Repair, Ghent University, B-9000 Ghent, Belgium; Leen.VandeSande@UGent.be (L.V.d.S.); Wim.Ceelen@ugent.be (W.P.C.); 3Department of General Surgery, University Hospital of Wales, Cardiff CF14 4XW, UK; jared.torkington@wales.nhs.uk

**Keywords:** oncolytic virus, delivery, peritoneum, PIPAC, PIPAV, peritoneal carcinoma, pressurised, aerosolised, metastasis

## Abstract

Background: The prognosis of patients with peritoneal metastases is poor. Treatment options are limited because systemically delivered chemotherapy is not usually effective in this type of disease. Pressurised intraperitoneal aerosolised chemotherapy (PIPAC) is a recently developed alternative technology for delivering intraperitoneal chemotherapy, potentially enhancing treatment efficacy. Here, we assess the feasibility of pressurised intraperitoneal aerosolised virotherapy (PIPAV) to deliver a different class of anticancer agents, oncolytic adenoviruses, in vitro and in vivo. Methods: Adenoviral vectors expressing reporter genes green fluorescence protein (Ad5.GFP) or firefly luciferase (Ad5.Luc) were subject to pressurised aerosolisation. The ability of the virus to survive PIPAV was assessed in vitro and in vivo by monitoring reporter gene activity. Wistar rats subjected to PIPAV were assessed for any adverse procedure related events. Results: In vitro transduction assays demonstrated that Ad5 retained viability following pressurised aerosolisation and could transduce permissive cells equally effectively as non-aerosolised control vector. PIPAV was well tolerated in rats, although minimal transduction was observed following intraperitoneal administration. Conclusions: PIPAV appears viable and well tolerated, though in vivo efficacy requires further optimisation.

## 1. Introduction

Malignant cells from an intra-abdominal primary tumour may invade the serosal lining of the abdominal cavity causing peritoneal metastases (PM). PM are a late feature of many cancers, making them common. They are challenging to treat, and prognosis is generally poor. For example, the reported 5-year survival rate for patients with isolated peritoneal metastases from colorectal cancer treated with systemic chemotherapy ranges from 0 to 19% [1]. In patients with stage III ovarian cancer (i.e., spread to the abdominal cavity) the average 5 year survival rate reported is 39% [1]. Thus, peritoneal metastases represent an area of significant unmet clinical need.

A number of studies have demonstrated that patients with isolated peritoneal metastases have poorer survival outcomes than patients who have isolated metastases in solid organs, such as the liver or lungs, if both groups are treated with systemic chemotherapy [2]. It is hypothesised that this is because the anatomical and physiological features of the peritoneal mesothelium render it relatively resistant to systemically administered therapeutics [3]. The capillary network supplying the mesothelial surface is located in a submesothelial connective tissue layer, and it has been suggested that this acts as a ‘barrier’ to drug transport [3,4,5].

Intraperitoneal drug delivery has been suggested as a potential solution to this problem. Pressurised intraperitoneal aerosol chemotherapy (PIPAC) is a recent innovation for intraperitoneal administration of chemotherapy to treat peritoneal disease in advanced malignancy [6]. During laparoscopic (keyhole) surgery, the abdominal cavity is inflated with carbon dioxide, creating a pneumoperitoneum of 12 mmHg, to allow the surgeon space to work. PIPAC involves aerosolising chemotherapy solutions into the abdominal cavity once the pneumoperitoneum is established using a specially designed laparoscopic nebuliser (Capnopen, Capnomed, Zimmern ob Rottweil, Germany). The aerosol is then allowed to circulate in the pneumoperitoneum, which is maintained for 30 min. The hypothesis behind PIPAC is that the pressure of the pneumoperitoneum counteracts the tumour interstitial pressure, increasing absorption and penetration of drugs administered into the abdominal cavity. Pre-clinical studies in animal models have demonstrated that the distribution of solutions delivered using PIPAC is extensive and may be superior to simple lavage of the cavity [7]. In vitro testing using human peritoneum demonstrated increased depth of penetration of a fluorescent-labelled therapeutic agent applied using PIPAC compared to the same agent applied using lavage [8]. The technique has been used extensively to deliver chemotherapy in phase I/II trials, and the administration of advanced therapeutics is now starting to be assessed in vitro and in vivo [9]. Thus far, the delivery of nanoparticles [10] and messenger RNA [11] have been reported.

Adenoviruses have long been identified as having the potential to be effective anti-cancer agents. The structure and biology of human adenovirus serotype 5 (Ad5) has been well described and understood [12]. Like other adenoviruses, they are non-enveloped, with a double stranded DNA genome contained within an icosahedral capsid. They are obligate intracellular parasites and can infect both dividing and non-dividing cells. The primary receptor for Ad5 is the coxsackie and adenovirus receptor (CAR) [13]. They are lytic viruses, causing destruction of the host cell on infection and release of progeny viruses into the tissue. Thus, the therapeutic effect of an oncolytic virus can be amplified, since further cell killing by progeny takes place following the cell killing from the initial infection of tumour cells. Furthermore, it has been shown that the Ad5 genome can be manipulated by using simple recombinant DNA techniques, to achieve high levels expression of foreign DNA inserts [12]. High yields of virus can be generated and collected [12]. These properties are all advantageous for an adenovirus-based cancer treatment and Ad5 has been a popular vector choice for groups developing biological therapies as a result [14].

However, a potential limitation of Ad5 based vectors is that existing immunity is common because of high seroprevalence rates across the world [15,16]. Studies have shown that the humoral response to Ad infection generates antibodies targeted at epitopes on multiple domains of the capsid. These antibodies act synergistically to neutralise the virus on re-exposure [17]. This means that vectors may be neutralised by antibodies before they reach their target tissue. Furthermore, it has been demonstrated that Ad5 forms complexes with coagulation factor X in vivo to enter cells using heparan sulphate proteoglycans (HSPGs), abundant on hepatocytes, as well as other cell types [18,19]. This results in rapid sequestration of systemically administered Ad5 vectors by the liver and may impact on uptake to other off target tissues following other routes of delivery.

One method of counteracting virus neutralisation is to administer the vector directly into the target tissue, therefore enabling rapid entry of the virus to target cells and circumventing components of humoral immunity. The PIPAC technique potentially represents a useful delivery method to enable delivery of oncolytic viruses directly to peritoneal metastases in the abdominal cavity. Intraperitoneal delivery would avoid interactions with blood proteins and circulating antibodies that would occur as an intravenously administered vector was delivered to the peritoneum in the blood.

One problem with direct administration to peritoneal metastases is that patients with abdominal cancers frequently have ascites present in the peritoneal cavity, and this fluid has also been shown to contain antibodies and immune cells that can neutralise adenovirus vectors [20,21]. However, the ascites is removed at the start of a PIPAC procedure. Thus, delivery by pressurised intraperitoneal aerosolisation may enable the provision of a more favourable environment for the virus to bind and enter target cells. Our aim was therefore to assess the feasibility of pressurised intraperitoneal aerosolisation as a potential method for the delivery of oncolytic adenovirus therapies to treat peritoneal metastases.

## 2. Materials and Methods

### 2.1. Viruses

The vectors used were based on an Ad5 genome captured in a bacterial artificial chromosome (BAC). Subsequent modifications were introduced into the BACs by homologous recombineering [22]. Ad5 was engineered to express Green Fluorescent Protein (GFP) as a reporter gene under the control of the short CMV IE promoter (Ad5.GFP), or to express Luciferase as a reporter gene by the same method (Ad5.Luc). They had a complete *E1/E3* gene deletion, rendering them replication deficient. Viruses were produced in HEK293 cells and purified and characterised according to standard protocols [23].

### 2.2. Cell Lines

A Wistar rat cell line (CC1, ECACC 93070901) was purchased from Public Health England (Salisbury, UK). Chinese Hamster Ovarian Epithelial (CHO-K1) cells and Chinese Hamster Ovarian transfected to express human CAR (CHO-CAR) were gifted by Dr Lynda Coughlan.

### 2.3. In Vitro Assays

#### 2.3.1. Effect of Aerosolisation

Cell lines were cultured and detached from the flasks when sub-confluent using 0.05% trypsin. Complete medium was added to neutralise the trypsin, and then the cells were counted using a haemocytometer. Cells were seeded in a flat bottomed 96 well plates at 20,000 cells per well and incubated overnight at 37 °C and 5% CO_2_.

Adenovirus vectors had been previously manufactured by homologous AdZ recombineering [22]. The Ad5.Luc and Ad5.GFP viruses were diluted in serum-free media (specific to cell type) to a concentration of 2 × 10^9^ vp/mL. Half of each solution was reserved to carry out the control experiments. The remaining solutions were aerosolised using a High-Pressure Injector (HPI) (Mark 7 Arterion Pedestal, Beyer, Leverkusen, Germany) connected to a CapnoPen™ aerosoliser (Capnomed GmbH, Zimmern, Germany). The aerosolised samples were collected in a clean 50 mL falcon tube.

The medium was removed from the wells of the 96 well plates and the cells were washed with 200 µL PBS. The virus solutions were then diluted with appropriate serum-free media and added to the wells, with four replicates per condition (aerosolised and non-aerosolized). A total of 100 µL of serum free medium was used as a negative control, with four replicates per cell line. The plates were returned to the incubator for 3 h. The medium was then removed from each well, the cells were washed with 200 µL of PBS, and complete medium was added. The cells were incubated for a further 45 h. Expression of the reporter genes was then assessed.

The expression of the luciferase reporter gene was assessed by Luciferase Assay System (#1501, Promega UK Ltd., Southampton, UK) as per the manufacturer’s protocol. The medium was removed from the wells and the cells were washed with PBS. Lysis reagent (#E1531, Promega UK Ltd., Southampton, UK) was diluted in water as per the manufacturer’s instructions and 100 µL was added to each well. The plates were frozen at −80 °C. The plates were thawed, the contents of the wells mixed, and then 20 µL of the lysate were transferred to a white 96 well plate. Then, 100 µL of luciferase reagent were added to each well just prior to measurement of the luciferase activity in Relative Light Units (RLU) on a multimode plate reader (Clariostar, BMG Labtech, Ortenberg, Germany). The RLU was then normalised for total cellular protein (RLU/mg). The protein concentration in each well was determined using a bicinchoninic acid (BCA) protein assay (#23227; Pierce™ BCA Protein Assay Kit, Thermo Scientific, Loughborough, UK) as per the manufacturer’s instructions. Bovine serum albumin (BSA) was used as a protein standard at concentrations 2.0, 1.5, 1.0, 0.75, 0.5, 0.25, 0.125, 0.025, and 0 mg/mL in PBS. Next, 10 µL of each standard, and 10 µL of each sample, were transferred to a 96 well plate in duplicate. A total of 200 µL of working reagent (A:B = 50:1) was then added to each well, and the plates were incubated for 30 min. The absorbance was measured at λ570 nm on an iMark™ Microplate Absorbance Reader (BioRad, Hertfordshire, UK). The optical density (OD) was normalised in all wells by subtracting the 0 mg/mL (PBS only) value from the reading. A standard curve was prepared using the OD obtained from the BSA standards, and the protein concentration in each well was deduced from the equation of the standard curve. A two-way ANOVA with Tukey’s multiple comparison test was used to compare the luciferase expression in each cell type and between those infected with aerosolised and non-aerosolised virus.

The effect of different pressure conditions on the viability of Ad5.Luc was also assessed. Plates were prepared in the same way, with CHO K1, CHO CAR, and CC1 cells plated at 20,000 cells per well and incubated for 24 h in complete medium. A solution of Ad5.Luc in serum-free medium was made and serial dilutions carried out prior to application to the cells in triplicate. The plates were placed in hyperbaric apparatus and the pressure was maintained for 30 min. After 30 min, all plates were put in the incubator for a further 1 ½ hours. The virus solution was removed and replaced with complete medium. After a further 48 h incubation, the medium was removed and the cells were washed and frozen in lysis buffer. The protein concentration in each well was determined using a BCA assay, and the expression of luciferase by luminometry after the addition of luciferin as described above. The mean RLU detected was normalised to protein concentration. A two-way ANOVA with Tukey’s multiple comparison test was performed in GraphPad Prism 9.0 to compare the results between the different pressure conditions.

The expression of GFP was assessed using flow cytometry. To prepare the cells, the medium was removed from each well and the cells were washed twice with PBS. Trypsin was then added to each well and the cells were incubated for 10 min. The trypsin was neutralised by the addition of complete medium to each well, and then after mixing the cells were transferred to a round-bottomed 96 well plate. The plate was spun at 1500 rpm for 5 min. The supernatant was removed, and the cells were re-suspended in 100 µL of 2% paraformaldehyde (PFA). The plate was left on ice for 15 min.

Once fixed, the samples were assessed on a BD Accuri C6 (BD Biosciences, Franklin Lakes, NJ, USA) flow cytometer. A total of 10,000 events were recorded in channel FL-1. Flow cytometry data were analysed using FlowJo v10 (FlowJo, BD Life Sciences, Ashland, OR, USA). The negative control samples, which had only complete medium and no virus applied, were used to gate the virus-exposed samples. The percentage of cells expressing GFP was calculated, and multiple *t*-tests (GraphPad Prism, GraphPad Software Inc., La Jolla, CA, USA) were used to compare the aerosolised samples with the non-aerosolised samples.

#### 2.3.2. In Vivo Studies

In vivo experiments were performed at the Laboratory of Experimental Surgery (Ghent University, Ghent, Belgium). A rat model of PIPAC had already been developed and used to test other therapeutics [24]. A series of small pilot experiments were designed to assess the feasibility and tolerability of using the model to compare intraperitoneal injection of adenovirus vectors to intraperitoneal aerosolisation. The initial dose of Ad5.Luc to be tested was selected based on previous experiments in rodent models. All animal experiments were performed under approved protocols by the Animal Ethics Committee of the Faculty of Medicine and Health Sciences, Ghent University, Belgium (ECD 17–109 and ECD 18–23), and in compliance with Belgian Council for Laboratory Animal Science (BCLAS) guidelines for the Care and Use of Laboratory Animals.

Male Wistar rats (Envigo, Horst, The Netherlands) were used. The rats were allowed to acclimatise to the surroundings for at least 48 h and were kept in standard housing conditions with water and food at libitum and a 12 h light/dark cycle. The rats were housed grouped by treatment allocation.

In the first experiment, virus administration by intraperitoneal injection (group A) was compared with intraperitoneal aerosolisation (group B). A dose of 3 × 10^10^ Ad5.luc viral particles in 5 mL warmed 0.9% sodium chloride solution was used in both groups (n = 3 per group), with negative controls (group C) receiving 0.9% sodium chloride solution by intraperitoneal injection (n = 1) and aerosolisation (n = 1). The second experiment was a dose escalation protocol. Intraperitoneal injection (group D) was compared with intravenous injection (group E) in the first part of the study. A dose of 1 × 10^11^ vp Ad5.Luc in 5 mL of warmed 0.9% sodium chloride solution was used in both groups (n = 1 per group). Intraperitoneal injection of 0.9% sodium chloride solution was used as a control (n = 1) (group F).

After intervention, the rats were observed for 72 h. Welfare observations were made daily based on the rat grimace scale and body weight, and analgesia (Ketoprofen, 5 mg/kg, SC) was provided as per the Animal House protocol. The rats were then imaged to determine the distribution of the Ad5.Luc by in vivo luminometry (IVIS Lumina II, PerkinElmer, Zaventem, Belgium). Rats were anesthetised by inhalation of sevoflurane and injected with 1.5 mL of D-luciferin (PerkinElmer, Zaventem, Belgium) at 15 mg/mL into the peritoneal cavity. Images were captured at 10 min. A one minute exposure was used in groups A, B, and C, and a 10 min exposure was used in groups D, E, and F. Images were analysed by Living Image^®^ software (PerkinElmer, Zaventem, Belgium) on a normalised scale (photons/second/cm^2^/steradian). The rats were sacrificed with a lethal injection of T-61 (0.3 mL/kg, IV) into the tail vein and ex vivo luminometry of individual organs and tissues was performed.

### 2.4. Statistical Analyses

Figures were created and statistical analyses performed with GraphPad Prism version 9.0 (GraphPad Software Inc., La Jolla, CA, USA). Unless otherwise stated, data show the mean ± standard deviation (SD) or standard error of the mean (SEM) of *n* = 3–4 (specific *n* numbers are indicated in each figure legend). *p* values were as follows: ns = not statistically significant (*p* > 0.05); * = *p* < 0.05; ** = *p* < 0.01; *** *= p* < 0.001; **** = *p* < 0.0001.

## 3. Results

### 3.1. Adenovirus Vectors Survive Aerosolisation and Retain Their Ability to Transduce Cells Expressing Their Native Receptor In Vitro

To determine whether aerosolisation of adenoviral vectors using the CapnoPen device negatively impacted on viral fitness, we performed transduction assays using aerosolised and non-aerosolised Ad5.GFP (Figure 1). For these studies we utilised CHO CAR cells, which express CAR, the native receptor for Ad5, and CHO K1, a cell line with no expression of CAR, used for comparison. Cell receptor expression was confirmed using flow cytometry (Figure 1a). Following infection of the CHO cells with aerosolised and non-aerosolised samples of Ad5.GFP, flow cytometry was carried out to compare the number of cells expressing GFP from the non-aerosolised virus experiments versus the aerosolised virus experiments. Gating for the flow cytometry was carried out using the control cell populations, which were not exposed to virus, but maintained in complete medium. The equipment used to aerosolise the virus solution is shown (Figure 1b), and flow cytometry data is shown (Figure 1c). The results from the non-aerosolised virus experiments demonstrate that, as expected, there was minimal expression of GFP in the transduced CHO K1 cells due to the lack of CAR. The non-aerosolised Ad5 GFP was able to transduce CHO CAR cells in a dose-dependent fashion. The same pattern of GFP expression was observed in samples infected with aerosolised virus, suggesting that aerosolisation has no intrinsic effect on the virus fitness. The proportion of CHO CAR cells expressing GFP in the non-aerosolised versus the aerosolised samples was compared (Figure 1d) and there was no significant difference in expression of GFP in cells inoculated with aerosolised virus when compared to cells inoculated with non-aerosolised virus at any viral titre (1000 vp/cell *p* = 0.47, 2500 vp/cell *p* = 0.15, 5000 vp/cell *p* = 0.46).

### 3.2. The Development of a Rat Model to Test Pressurised Intraperitoneal Aerosolisation of Adenovirus Vectors

#### 3.2.1. Adenovirus Vectors Are Able to Transduce Wistar Rat Hepatocytes In Vitro after Aerosolisation Using the CapnoPen Device

An Ad5 vector with a firefly luciferase reporter gene (Ad5.Luc) was selected as the best suited to an exploratory in vivo study, as it would enable luminometry by In Vivo Imaging System (IVIS) (IVIS Lumina II, PerkinElmer, Zaventem, Belgium) imaging to be used assess the expression of the reporter gene in the live animals. A preliminary experiment was therefore performed to test the ability of Ad5.Luc to transduce rat hepatocyte CC1 cells in vitro after aerosolisation using the CapnoPen™ device (Capnomed GmbH, Zimmern, Germany). CHO CAR and CHO K1 cells were again used as positive and negative controls.

The relative expression of luciferase in each cell line after transduction with aerosolised and non-aerosolised virus was quantified (Figure 2a). There was no significant difference in the RLU detected from cells incubated with aerosolised virus at a concentration of 10,000 vp/cell compared to those incubated with non-aerosolised virus at the same concentration. There was a significant difference in the mean RLU detected in the CC1 samples compared to the CHO K1 samples, suggesting that the Ad5.Luc was able to infect and transduce these cells, though not as efficiently as the CHO CAR cells.

#### 3.2.2. Adenovirus Vectors Are Able to Transduce Wistar Rat Hepatocytes In Vitro under Atmospheric and Hyperbaric Conditions

Since delivery of therapeutics using pressurised intraperitoneal aerosolisation is performed under increased pressure, we sought to ascertain whether adenoviral vectors retained complete ability to infect cells when placed under supraphysiological pressures. We therefore assessed the ability of Ad5.Luc to transduce CC1 cells at a range of concentrations at atmospheric pressures or at hyperbaric pressures of 20 or 40 mmHg (Figure 2b). The incubation of the cells with the virus solution was carried out in a pressurised chamber. There was no significant difference observed between cells incubated at atmospheric pressure, 20 or 40 mmHg at any concentration. The pressures tested were higher than those that would be used in the Wistar Rat model in vivo (i.e., 8 mmHg), providing comfort that pressures used in vivo ought not to hamper adenoviral fitness.

#### 3.2.3. Intraperitoneal Aerosolisation of an Adenovirus Vector Is Feasible and Is Tolerated by Healthy Wistar Rats

Following the successful in vitro experiments, we next wished to test the feasibility of administration of virus vectors in an in vivo Wistar rat model. The pressurised aerosolisation procedure was completed in four healthy Wistar rats (Group B and C2). Following intervention, all rats were observed for 72 h. The rats who had undergone the aerosolisation procedure had higher welfare scores (indicating greater distress) and required analgesia on day 1 and day 2 post procedure. This was the case in both the rats receiving adenovirus via aerosolisation, and the rat that received saline. Rats receiving adenovirus by intravenous or intraperitoneal injection did not require analgesia. In vivo imaging using the IVIS system was carried out 72 h after administration to assess the level of transduction (Figure 3a). All the rats treated by intraperitoneal aerosolisation had 13 photons/second/cm^2^/steradian or less detected with no sign of localisation, indicating it was likely to represent background signal.

#### 3.2.4. Intraperitoneal Administration of an Adenovirus Vector Results in Detectable Transduction in a Wistar Rat Model

IVIS images were obtained after injection with the higher dose of virus (Figure 3b). There was a greater signal seen with intravenous injection compared to intraperitoneal injection, likely due to the high concentrations of blood clotting factor X in the blood which can be engaged by Ad5 based virotherapies to efficiently enter cells presenting HSPGs. The rats tolerated the increased dose well, with no increase in welfare scores and no requirement for analgesia.

## 4. Discussion

The feasibility of using pressurised intraperitoneal aerosolisation to administer oncolytic adenoviruses was investigated by preliminary work in vitro and in vivo using replication deficient Ad5 vectors expressing reporter genes. The in vitro experiments demonstrate that aerosolisation using the CapnoPen device does not affect the ability of Ad5-based vectors to infect and transduce cells. Additionally, the virus is unaffected by exposure to hyperbaric conditions, and at pressures which exceeded those used clinically during PIPAC procedures was able to transduce cells efficiently. These findings were expected, since Ad5 causes respiratory illness and has therefore evolved to survive transmission by respiratory droplets. However, these are important findings in determining whether intraperitoneal aerosolisation will be of use as a method of delivery for oncolytic adenoviruses. Two Ad5 vectors were tested here; Ad5.GFP and Ad5.Luc. These differ from vectors that would be used in a clinical context since they are non-replicating and have relatively few modifications. This is a limitation of this study since these are not vectors that would ultimately be used in the clinic. However, it is anticipated that more highly engineered vectors based on Ad5 would tolerate aerosolisation equally well. One such vector has been developed and described by Uusi-Kerttula et al. [23,25]. Ad5_NULL_A20 is an Ad5 vector that has native tropisms ablated and has been retargeted to the tumour-selective integrin αvβ6 through incorporation of an αvβ6-binding peptide (A20, NAVPNLRGDLQVLAQKVART) within the fibre knob domain HI loop. This vector has been shown to specifically infect epithelial ovarian cancer cells in vitro [23] and in vivo [21]. Further investigation to test the ability of the Ad5_null_A20 vector, and other specialised vectors, to survive aerosolisation and retain their ability to specifically infect and kill target cells would be an important area for future assessment of the feasibility of this technique in clinical practice.

Administration to the local tumour environment using a technique such as PIPAV may still be of benefit even when using a highly targeted vector because such vectors can still be affected by neutralisation by the immune system. As already mentioned, the ascitic fluid of patients previously exposed to adenoviruses can contain immune cells and antibodies capable of neutralising a therapeutic vector [20,21]. The surgical part of the PIPAV procedure in a human subject would allow drainage of the ascites, as well as peritoneal lavage with warmed saline prior to administration of the therapeutic agent, thus effectively removing and diluting factors which may negatively impact the efficacy of oncolytic adenovirus. The direct administration of the vector to the peritoneal surface using the nebuliser would potentially increase the concentration of the vector in direct contact with the target tissue, and therefore the likelihood of virus binding and endocytosis occurring before sequestration or neutralisation. It may also allow a lower dose overall to be used, which might have positive implications for the safety and cost of treatment. Further studies of adenovirus vectors in development using 3D tissue culture, and in the presence of varying concentrations of ascites, would be useful to understand whether greater tumour lysis occurs when ascites is removed in this way.

To ascertain whether the Wistar rat model would be an appropriate way to test intraperitoneal aerosolisation of adenovirus vectors, experiments were performed in vitro using a Wistar rat hepatocyte cell line (CC1). The Ad5 vector best suited for the rat model, Ad5.Luc was able to transduce the Wistar rat cells in vitro with reasonable efficiency, demonstrating that the rat cell line was able to be infected with a human adenovirus and express a transgene. This did not translate into the expected results in the pilot experiment in vivo using a dose of 3 × 10^10^ vp of Ad5.Luc. The dose selected was conservative when compared with experiments previously carried out by the group in mice, but we were conscious that the rats were going to undergo the aerosolisation procedure as well as receiving the virus and did not want to induce dose-limiting toxicity. Additionally, we discussed the proposed dose with other groups who used rat models to assess adenovirus-based vectors. The rats tolerated the procedure and the administration of the vector well. However, transduction was not detected.

The second in vivo experiment therefore tested a higher dose of virus (1 × 10^11^ vp) and compared transgene expression after IV and intraperitoneal delivery. The intraperitoneal route did result in transduction, though at a lower rate than IV delivery. This is perhaps expected given previous research about the intraperitoneal route of administration in humans [26] and other animal models [27,28] has also shown a lower uptake of therapeutic agents. The same rat model has been used to compare intraperitoneal injection, intraperitoneal aerosolisation, and IV injection of mRNA complexes encoding the firefly luciferase protein and found no significant difference in the average overall luminescence detected from rats treated using each of the three administration routes [11]. The authors commented that intraperitoneal aerosolisation and IV injection produced less variability in the overall luminescence detected. There was a trend towards greater overall luminescence in the intraperitoneal injection group, but this was not significant because there was more variability between rats treated using that route of administration. When they looked at the distribution of the luminescence, the intraperitoneal aerosolisation technique seemed to produce luminescence from a greater surface area of the abdomen of the imaged rats, however, again, this was not a significant difference because of variability between rats. Intraperitoneal injection and IV injection produced much less variable distribution, with the signal concentrated to a smaller area in the upper abdomen. In this study using adenovirus vectors, the high luminescence in the upper abdomen following intravenous delivery is likely to represent transduction in the liver of the rat. Ad5 forms complexes with coagulation factor X to enter cells using heparin sulphate proteoglycans, and these are abundant on hepatocytes [18,19]. We would not expect to see this pronounced effect if an engineered vector was used as it is possible to ablate native tropisms of the virus to achieve greater uptake in target tissues [21].

The detection of transduction in the second in vivo study, albeit at a lower level from the intraperitoneal route of administration, supports further investigation in this rat model. The rats did not appear to suffer any adverse consequences from the increased dose. These results combined with the finding from the first in vivo study that the rats were able to tolerate the intraperitoneal aerosolisation procedure using the lower dose of Ad5.Luc support further investigation in this model. The absence of a dose escalation study in vivo is a limitation of our work to date. An amendment to our ethics approval to perform the second stage of the second in vivo study, comparing intraperitoneal aerosolisation and intraperitoneal injection with an increased dose of 3 × 10^11^ vp Ad5.Luc was submitted and approved in January 2020. Ultimately, the Severe Acute Respiratory Syndrome Coronavirus 2 pandemic prevented this experiment from going ahead as planned. This therefore remains a promising avenue for future investigation.

## 5. Conclusions

We demonstrate that aerosolisation and hyperbaric pressure do not impact on the fitness of adenoviruses, paving the possibility that such platforms could be amenable to pressurised intraperitoneal aerosolised virotherapy applications in the future for the treatment of peritoneal metastases. The in vivo feasibility study in Wistar rats demonstrated the viability and tolerability of this approach, however, further studies at higher doses are required to fully assess the efficacy of this approach for in vivo applications.

## Figures and Tables

**Figure 1 pharmaceutics-13-02043-f001:**
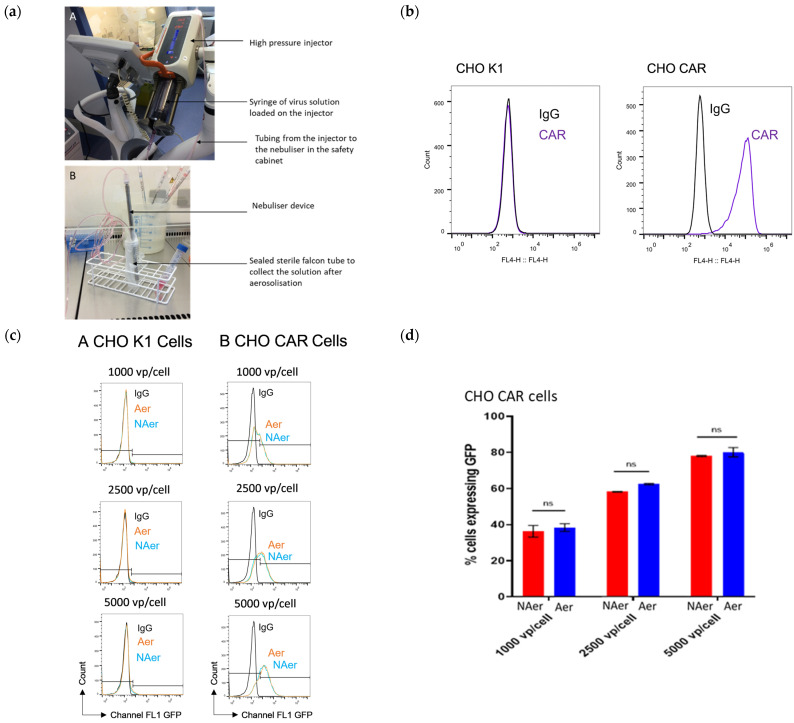
Ad5.GFP survives aerosolisation using the CapnoPen device and retains its ability to transduce Wistar Rat hepatocytes in vitro. (**a**) Flow cytometry data from immunocytochemistry to confirm receptor expression of CHO cells. (**b**) Pictures to show the equipment used for the aerosolisation of the virus solution in vitro. (**c**) Flow cytometry data to show the number of cells expressing GFP after infection with non-aerosolised (NAer) and aerosolised (Aer) vector. (**d**) Quantification of CHO CAR cells expressing GFP after infection with non-aerosolised (NAer) and aerosolised (Aer) vector. Error bars represent standard deviation; ns = not statistically significant (*p* > 0.05).

**Figure 2 pharmaceutics-13-02043-f002:**
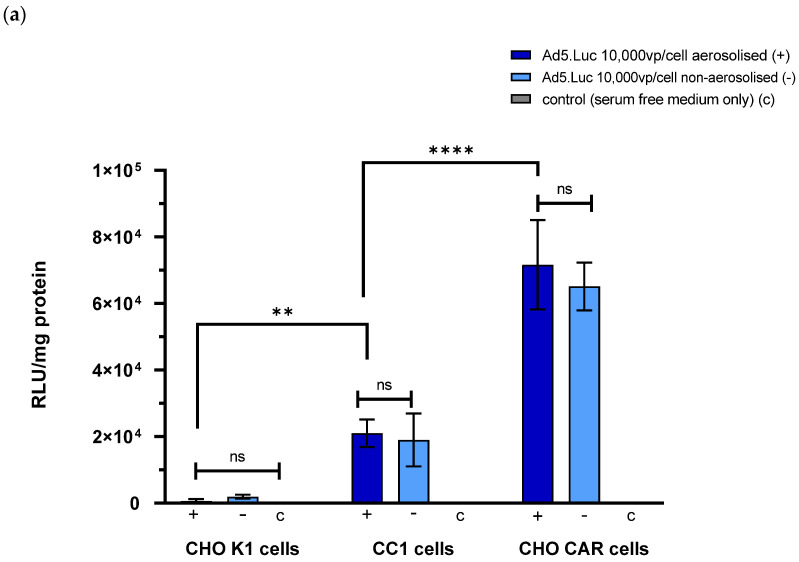
Ad5.Luc survives aerosolisation using the CapnoPen device and retains its ability to transduce Wistar Rat hepatocytes in vitro under hyperbaric pressures. Quantification of luciferase expression in CHO K1, CC1, and CHO CAR cells following transduction with aerosolised Ad5.Luc versus non-aerosolised Ad5.Luc versus control wells (**a**). Luciferase expression mediated by Ad5.Luc at varying concentrations at under varying hyperbaric pressures of 20 mmHg; ns = not statistically significant (*p* > 0.05); ** = *p* < 0.01; **** = *p* < 0.0001. (**b**). Error bars represent standard deviation.

**Figure 3 pharmaceutics-13-02043-f003:**
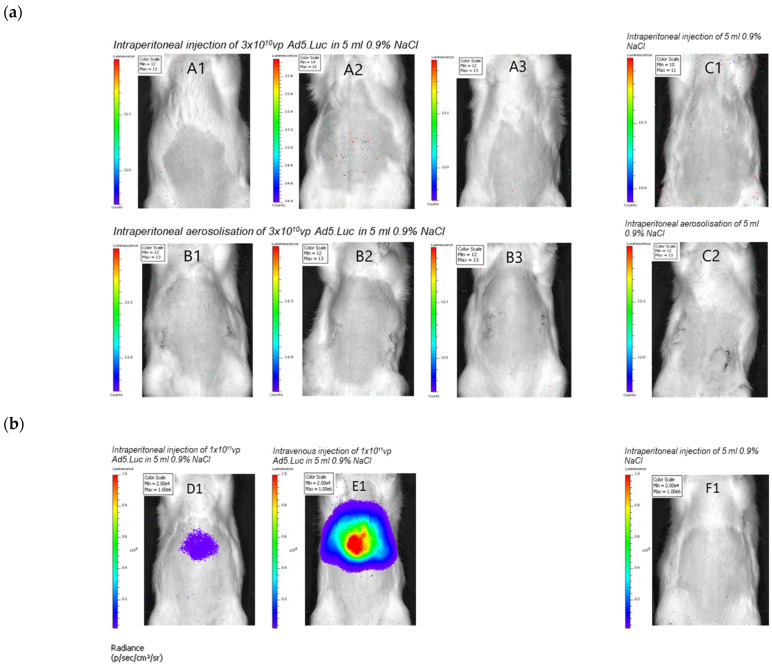
Intraperitoneal administration of an adenovirus vector results in detectable transduction in a Wistar Rat model and is well tolerated. IVIS imaging of rats at 72 h after intervention to determine the adenovirus vector biodistribution after an intervention dose of Ad5.Luc 3 × 10^10^ vp (**a**). IVIS imaging of rats at 72 h after intervention to determine the adenovirus vector biodistribution after an intervention dose of Ad5.Luc 1 × 10^11^ vp (**b**). Rat weights following intervention (**c**). Weight loss >0 but ≤10% prompted administration of analgesia.

## Data Availability

All data presented in this study are available within the manuscript.

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
