# Peer review of "The Feasibility of Pressurised Intraperitoneal Aerosolised Virotherapy (PIPAV) to Administer Oncolytic Adenoviruses"

_pharmaceutics, 2021, doi:10.3390/pharmaceutics13122043_

Round 1

Reviewer 1 Report

The findings of this study are of little clinical relevance and thus of limited interest to the readers of Pharmaceutics.

Author Response

The findings of this study are of little clinical relevance and thus of limited interest to the readers of Pharmaceutics.

Author Response: We thank the reviewer for taking the time to read our manuscript and make their assessment. We saddened by the brevity of their review and lack of constructive critique or suggestions on how we might improve the article. We remind the reviewer that the editorial team have previously assessed the manuscript and deemed it worthy of review, and that this special edition of Pharmaceuticals is dedicated to the “Advanced Use of Adenoviral Vectors in Cancer and Gene Therapy”. Our article fits squarely within the remit of this special edition. We naturally disagree with the reviewer’s opinion that the “findings of the study are of little clinical interest”. We are grateful that the remaining reviewers provided more thoughtful, insightful, and supportive comments, saw significant clinical value in our manuscript and were positive in supporting its publication subject to certain suggested alterations. Since reviewer one provided no suggestions for improvement, we have not made any alterations to the manuscript in response. We are saddened by their review and the disservice it does to the peer review process in general, and glad for the opportunity to make this review and response publicly available for others to view.

Reviewer 2 Report

The manuscript represents a very important approach to tumour therapy after surgery in the abdominal area by the use of pressurized virotherapy. The procedure, however, can be used for other events, ie intraperitoneal infections making the proper adjustments. The approach is interesting and well designed. The experiments are well defined and the results are promising. There is only an issue to check if in figure 2 part b the concentration of 1250 a major difference is observed with 20 mm Hg In addition I would suggest to modify the colors of the bars.

Perhaps the authors may address in the discussion further details of virus deliveries and discussing very low to low-grade tumour metastasis in the abdominal area. 

The manuscript is well written and only minor grammatical/spelling mistakes were found.

Author Response

The manuscript represents a very important approach to tumour therapy after surgery in the abdominal area by the use of pressurized virotherapy. The procedure, however, can be used for other events, ie intraperitoneal infections making the proper adjustments. The approach is interesting and well designed. The experiments are well defined and the results are promising. There is only an issue to check if in figure 2 part b the concentration of 1250 a major difference is observed with 20 mm Hg In addition I would suggest to modify the colors of the bars.

Author Response: We thank the reviewer for their positive critique of our manuscript and for recognising the importance of our approach, and for describing our study as “interesting and well designed”. Regarding the specific comments, we have checked the statistics for figure 2b and a 2 way ANOVA with Tukey’s multiple comparisons test did not demonstrate any significant difference between the values obtained for the 1250 concentration at 0 vs 20 or 20 vs 40 mmHg. We have adjusted the colour scheme of the figure to make it more aesthetically pleasing.

Perhaps the authors may address in the discussion further details of virus deliveries and discussing very low to low-grade tumour metastasis in the abdominal area. 

Author Response: We feel that discussion of specific clinical scenarios is outside the scope of this in vivo feasibility study.  We have covered the broad clinical application of the technique, but the utility in any given cancer would be dependent on the development of an appropriate vector as much as the use of the PIPAV technique, and therefore we do not feel it is appropriate to extend the discussion further here.

The manuscript is well written and only minor grammatical/spelling mistakes were found.

Author Response: We thank the reviewer for this comment and have thoroughly checked the resubmitted version for grammatical and spelling mistakes.

Reviewer 3 Report

This is a manuscript describing the feasibility of utilizing pressurized intraperitoneal aerosolized virotherapy (PIPAV) to deliver oncolytic adenoviruses.

Dear authors,

I enjoyed reading your manuscript and the elegant studies that you performed.

I may have missed it, but I am still a bit confused as to why you did not test the higher dose of virus via intraperitoneal aerosolization in the second in vivo studies. Was this decision made after discussion with other groups? 

Was there any consideration to viral recovery or IHC of the peritoneum to further assess viral distribution?

And was there consideration to inoculating the rats with the ovarian cancer cell line to create a carcinomatosis model to compare delivery methods and assess transduction?

Kindest regards,

Author Response

 Author Response: We thank the reviewer for reviewing our manuscript. Unfortunately their review arrived literally as I was uploading the revised manuscript, the day before our revisions were due. We apologise therefore for only being able to respond briefly to their suggestions here.

I enjoyed reading your manuscript and the elegant studies that you performed.

Author Response: We thank the reviewer for their kind comment and positive assessment of our manuscript.

I may have missed it, but I am still a bit confused as to why you did not test the higher dose of virus via intraperitoneal aerosolization in the second in vivo studies. Was this decision made after discussion with other groups? 

Author Response: This is a very reasonable point and is covered in response to others. Unfortunately, we were unable to carry out the experiments due to the limitations placed on us by the COVID pandemic (access to lab to prepare virus, travel to Belgium to perform collaborative studies, key team member classified as Extremely Clinically Vulnerable and therefore subject to Government advice not to attend workplace). During this delay, the in vivo scientist who was trained to perform the studies moved on to another position. We have been sure to clarify in the modified discussion and additional study limitations section (as suggested by reviewer 5) that higher doses could and should be performed in future to further assess feasibility here

Was there any consideration to viral recovery or IHC of the peritoneum to further assess viral distribution?

Author Response: This was considered. Samples were shipped from Ghent (where in vivo studies were performed) to Cardiff for assessment, however they were delayed at customs, thawed, and destroyed. Blame for this lies squarely with BREXIT and the additional bureaucracy this inflicted on the UK where import and export of precious samples and reagents has been disregarded in the name of so-called sovereignty.

And was there consideration to inoculating the rats with the ovarian cancer cell line to create a carcinomatosis model to compare delivery methods and assess transduction?

Author Response: We thank the reviewer for this astute comment. This will be an important aspect of future research which we are developing collaboratively.

Reviewer 4 Report

This manuscript describes the effect of pressurized aerosolization on the infectivity and activity of adenoviral vectors. This study has high significance since this promotes the use of oncolytic adenoviral vectors to be used for intraperitoneal aerosolized virotherapy for treatment of peritoneal metastasis. The methodology is adequate and the results in general support the conclusions. I have outlined below some specific comments and suggestions.

1) The authors reason that the direct delivery of oncolytic viruses into the intraperitoneal cavity might protect them from neutralization. More proof needs to be given in this regard to substantiate these claims. Intraperitoneal cavities are normally highly abundant in macrophages and immune cells.  Are there any literature evidences that support these claims. They should be mentioned in the main text.

2) Since the subject of the manuscript involves intraperitoneal tumors, does the presence of ascites affect oncolytic viral delivery? This is especially true for ovarian cancer peritoneal metastasis that is characterized by abundant ascites levels. The authors should use a intraperitoneal ascites model to substantiate their claims.

3) More substantial reasons need to be given to explain why the IVIS signal was greater when the viruses were delivered i.v. as aopposed ti i.p. administration.

4) Fig. 3a seems to be unnecessay and may be removed as long as the results are mentioned in the text. Alternatively they may be placed in supplementary figures.

5) The levels of transduction may still be increased by further increasing the viral load in vivo. Have the authors tried this experiment ordo they think this may lead to unacceptable toxicity? 

Author Response

This manuscript describes the effect of pressurized aerosolization on the infectivity and activity of adenoviral vectors. This study has high significance since this promotes the use of oncolytic adenoviral vectors to be used for intraperitoneal aerosolized virotherapy for treatment of peritoneal metastasis. The methodology is adequate and the results in general support the conclusions. I have outlined below some specific comments and suggestions.

Author Response: We thank the reviewer for their precis of our manuscript and for stating that this study has “high significance”. We thank the reviewer for highlighting the below for our attention, which have addressed where possible to improve the manuscript.

1) The authors reason that the direct delivery of oncolytic viruses into the intraperitoneal cavity might protect them from neutralization. More proof needs to be given in this regard to substantiate these claims. Intraperitoneal cavities are normally highly abundant in macrophages and immune cells.  Are there any literature evidences that support these claims. They should be mentioned in the main text.

Author Response: This is an important point; we thank the reviewer for allowing us the opportunity to clarify this and modify our manuscript accordingly. The rationale for this statement is that prior to PIPAC/PIPAV, the ascites, which is indeed rich in immune cells as well as potentially neutralising antibodies, would be drained prior to insufflation of the abdominal cavity with carbon dioxide and PIPAC/PIPAV procedure. This draining of ascites and insufflation of the abdomen prior to PIPAC is critical as it allows a gas filled space to be formed in the abdomen, into which the aerosolised chemotherapy or virotherapy can be introduced. This removal of ascites prior to the procedure has two effects – (1) it significantly dilutes the immune cells and antibodies present which might otherwise neutralise an introduced virotherapy, and (2) it enhances the possibility that the introduced aerosolised chemotherapy/virotherapy will rapidly and directly contact target cells (ie peritoneal metastases) before immune cells have the opportunity to neutralise the therapy. We have clarified these points in the revised manuscript (see lines 90-110 and 386-403)

2) Since the subject of the manuscript involves intraperitoneal tumors, does the presence of ascites affect oncolytic viral delivery? This is especially true for ovarian cancer peritoneal metastasis that is characterized by abundant ascites levels. The authors should use an intraperitoneal ascites model to substantiate their claims.

Author Response: The reviewer is correct that ascites does affect oncolytic viral delivery, and in fact we have studied this in some of our previous publications (Uusi-Kerttula et al, Hum Gene Ther, 2015; Uusi-Kerttula et al, Clinical Cancer Research, 2018; Hulin-Curtis et al, Oncotarget, 2018). As mentioned above, we hope these effects would be minimised in the clinical scenario where ascites is removed prior to insufflation and PIPAC/PIPAV procedure. In our (unpublished) experience, whilst the majority of ascites have neutralising activity, some ascites samples actively enhance viral infectivity. This likely reflects the differing composition of each individual ascites samples and the balance of factors which either neutralise/inhibit viral transduction (e.g. neutralising antibodies) or promote viral transduction (e.g. blood clotting factors such as FX). Given the limited time frame we have to modify and return our manuscript (7 days) we are unable to perform additional assays, instead we have added some additional sentences and citation to the discussion to cover these points (see lines 400-403). 

3) More substantial reasons need to be given to explain why the IVIS signal was greater when the viruses were delivered i.v. as opposed to i.p. administration.

Author Response: The enhanced transduction following intravenous delivery is likely to relate to the presence of higher concentration of factors which promote adenoviral transduction – most obviously blood clotting factor X (FX) in the blood. The Ad5: FX complex is well understood to promote uptake in hepatocytes. We have added a short sentence in both the revised results (lines 346-7) and discussion to speculate on this (lines 434-438)

4) Fig. 3a seems to be unnecessary and may be removed as long as the results are mentioned in the text. Alternatively they may be placed in supplementary figures.

Author Response: Although the data is somewhat negative, we feel it is important to present these findings here. We prefer therefore to leave these images and data within the paper, and hope the reviewer understands.

5) The levels of transduction may still be increased by further increasing the viral load in vivo. Have the authors tried this experiment ordo they think this may lead to unacceptable toxicity? 

Author Response: The reviewer is correct. Our plan had been to increase the dose and ethical approval was received to perform these studies at a higher dose of 3x1011 vp in January 2020. However, we were unable to carry out the experiments due to the limitations placed on us by the COVID pandemic (access to lab to prepare virus, travel to Belgium to perform collaborative studies, key team member classified as Extremely Clinically Vulnerable and therefore subject to Government advice not to attend workplace). During this delay, the in vivo scientist who was trained to perform the studies moved on to another position. We have been sure to clarify in the modified discussion and additional study limitations section (as suggested by reviewer 5) that higher doses could and should be performed in future to further assess feasibility here.

Reviewer 5 Report

I read with great interest the work of Tate et al. and perhaps it might represent a starting point for future studies. The article is well written and the authors describe their experiments thoroughly.

Minor issues that should be addressed:

  • As the authors mention in the title, this is feasibility study and perhaps this should be included in the conclusions.
  • also, a paragraph on study limitations should be included.

Author Response

I read with great interest the work of Tate et al. and perhaps it might represent a starting point for future studies. The article is well written and the authors describe their experiments thoroughly.

Author Response: We thank the reviewer for the positive comments regarding our manuscript and highlighting that the article is well written and thorough.

Minor issues that should be addressed:

As the authors mention in the title, this is feasibility study and perhaps this should be included in the conclusions.

Author Response: In line with this comment we have modified the conclusions to highlight that this is a feasibility study (see lines 458)

also, a paragraph on study limitations should be included.

Author Response: We agree that this is important to discuss the limitations of the work and have added statements in the discussion related to study limitations (see lines, 374-375, 447-453).